# Metamaterial-based passive analog processor for wireless vibration sensing
Dajun Zhang[1], Akhil Polamarasetty[2], Muhammad Osama Shahid[1], Bhuvana Krishnaswamy[1] & Chu Ma [1] ✉

Real-time, low-cost, and wireless mechanical vibration monitoring is necessary for industrial applications to track the operation status of equipment, environmental applications to proactively predict natural disasters, as well as day-to-day applications such as vital sign monitoring. Despite this urgent need, existing solutions, such as laser vibrometers, commercial Wi-Fi devices, and cameras, lack wide practical deployment due to their limited sensitivity and functionality. Here we proposed a fully passive, metamaterial-based vibration processing device, fabricated prototypes working at different frequencies ranging from 5 Hz to 285 Hz, and verified that the device can improve the sensitivity of wireless vibration measurement methods by more than ten times when attached to vibrating surfaces. Additionally, the device realizes an analog real-time vibration filtering/labeling effect, and the device also provides a platform for surface editing, which adds more functionalities to the current non-contact sensing systems. Finally, the working frequency of the device is widely adjustable over orders of magnitudes, broadening its applicability to different applications, such as structural health diagnosis, disaster warning, and vital signal monitoring.

Real-time mechanical vibration sensing is necessary to capture the rich information in a variety of industrial, healthcare, and natural environmental monitoring applications[1]. For example, vibration monitoring is critical for the prediction of natural disasters such as typhoons, earthquakes, and avalanche calamities, and for meteorological observation and geological survey[2]. Vibrations induced by human breathing and heartbeat are useful vital signals for health monitoring[3]. Vibration monitoring also provides in-situ and non-destructive tools for diagnosing the structural health of vehicles[4], industrial equipment[5], buildings, public infrastructures[6], and materials[7]. Furthermore, monitoring the resonances in ultra-high frequency mechanical vibration systems serves as an important mechanism for high-sensitivity nanoelectronics sensors[8–10]. This vast range of applications has motivated researchers to explore and develop various non-contact and contact vibration measurement techniques[11,12].

Contact-based methods typically attach sensors to the vibrating surface and hence render higher accuracy; vibration displacement or acceleration measurements using strain type[13,14], piezoelectric[15,16], or electrokinetic[17] sensors are some of the well-known approaches. However, their high accuracy comes at the cost of their need for on-board signal processing that in turn adds to the overall cost for infrastructure, including data acquisition, high quality data transmission, dedicated power supply[18]. Non-contact methods, on the other hand, are relatively less expensive as they allow for

measurement and acquisition of mechanical vibrations without attaching sensors to the vibrating surfaces. There are long existing non-contact approaches based on electromagnetic eddy current[19] and capacitance measurements[20] that measure close to the vibration surfaces, optical interference based approaches that require cumbersome and expensive laser vibrometry setups among others[21]. Recent methods have proposed the use of video image processing[18,22–24], or Doppler effect of wave signals (Wi-Fi[25]/radio frequency identification (RFID)[26]/Ultrasound[27]) as more flexible and low-cost alternatives to the above identified non-contact methods. They aim to capture vibration signals at a lower cost using low complexity algorithms, in turn broadening its application with the possibility of integrating vibration sensing with Internet of Things networks[28].

While the non-contact measurement technologies based on video image processing, or doppler effect of electromagnetic (EM) signals (Wi-Fi/RFID/Ultrasound) are promising due to their lower cost, there is still a long way to go for their practical deployments due to a key limitation: poor sensitivity, defined as the smallest vibration amplitude that can be detected. For instance, it is challenging for EM-based sensing system to detect a vibration with an amplitude much smaller than one wavelength of the electromagnetic wave. In the case of RFID, one wavelength would be around 30 cm; in the case of Wi-Fi, it would be between 6 cm and 12 cm; this limitation is because of the nearly undetectable phase change generated by

[1]Department of Electrical and Computer Engineering, University of Wisconsin–Madison, Madison, WI 53706, USA. [2]Department of Computer Sciences, University of Wisconsin–Madison, Madison, WI 53706, USA. ✉e-mail: chu.ma@wisc.edu

the vibration amplitude to be measured. Similarly, in camera-based vibration sensing, the sensitivity is decided by the number of photosensitive sensors on chip and numerical aperture. The higher the desired sensitivity, the higher is the hardware cost, contradicting the motivation of low-cost, non-contact measurement techniques.

In this work, we propose and develop a hybrid mechanical vibration monitoring method that offers a high sensitivity to non-contact systems using a passive device that amplifies the vibration of the surface of interest. The proposed passive device amplifies and processes the vibration observed, which is then captured by a low-cost, remote data acquisition system such as camera, ultrasound, Wi-Fi, or RFID devices based on the application. The flexible design of the passive device renders it easy to tune, making it a general-purpose amplification device for a variety of applications.

## Results and discussion
### Working principle of the hybrid monitoring system
Figure 1 illustrates our proposed system. The vibration processing device (Fig. 1 top left) is attached to the surface of interest (whose vibrations are to be monitored continuously). Its passive amplification makes it battery-free, eliminating the need for dedicated power supply that is typical of contact-based methods. The proposed device consists of a metamaterial layer formed by a mass-loaded membrane mounted on the top of a rigid shell support[29–32]. A mass-loaded membrane is widely used in metamaterials to produce a resonance frequency at which the wavelength is much larger than the membrane size[30]. Such a deep-subwavelength resonance endows the metamaterial interesting properties such as negative wave refraction[33] and enhanced wave absorption[34]. In our case, the mass-loaded membrane produces vibration amplification as well as gives our device a compact size and flexible deployment options, especially in lower frequency range (below tens of Hz). When the device is attached or fixed on to the surface of interest, its rigid shell will vibrate along with the subject (surface), which will cause the movement of the top membrane. When the monitored subject/surface vibrates with a frequency close to the eigenfrequency of the device, the device will be excited to a mechanical resonance state. As a result, the vibration magnitude of the membrane will be much higher than that of the monitored surface for this frequency. The resonance-induced vibration magnitude amplification is exploited to enhance the vibration measurement sensitivity; it also works as a physical layer label to help source identification in non-contact vibration measurement. In our paper, we use two low-cost data acquisition systems, a Wi-Fi device and a camera, to capture the vibration amplitude that is amplified by our passive mechanical device. In the case of Wi-Fi, our proposed device's amplification improves the doppler shift of electromagnetic signals, which in turn improves the sensitivity of the system, while in case of camera video, the amplification improves the frame-to-frame variance.

### Experimental characterization of the vibration processing device
As shown in Fig. 2a, the vibration amplification device is composed of a metamaterial layer mounted on the top surface of a rigid cylindrical shell. The metamaterial layer is composed of a membrane loaded by a mass at the center. The bottom of the shell is attached to the vibrating surface. The system can be characterized in a simplified form as the combination of a mass $m$ and a spring with stiffness $k$ determined by the membrane size, thickness, material, and strain. The influence of these parameters on the eigenfrequencies of the device is simulated in COMSOL and the results are shown in Supplementary Note 1. The vibration of the rigid cylinder follows the motion of the vibrating subject, which is the input of the device. The output is the vibration of the metamaterial layer. The metamaterial layer can be excited to resonance when the input vibration has a frequency close to its eigenfrequency. At this frequency, the vibration of the membrane will be largely enhanced compared to the vibration of the bottom surface.

In device characterization, the bottom of the device is attached to a rigid plate. The vibration is triggered by exciting a shaker (Bruel & Kjaer Type 4809) connected to the rigid plate. The vibration of both the bottom plate and the top membrane are measured with a laser

vibrometer (Polytec PSV-200 Scanning Laser Vibrometer System) for a range of frequencies. Figure 2b shows an example set of input and output vibration signals measured from a sample device with eigen-frequency of 9.5 Hz. When the input vibration has a frequency of 9.5 Hz and vibration amplitude of 0.76 mm, the output vibration amplitude reaches 5.3 mm, showing approximately 7-fold amplification. We define the peak-peak displacement of the membrane normalized by the peak-peak displacement of the bottom plate as the amplification coefficient. Figure 2c shows the experimentally measured amplification coefficient for two different devices having the eigen-frequency of 9.5 Hz and 110 Hz, respectively. For each device, there is a peak in the amplification coefficient vs. frequency curve near its eigenfrequency, with approximately 7-fold and 9-fold vibration amplitude amplifications, respectively. When the input frequency is smaller than the eigenfrequency, the amplification coefficient is approximately one, meaning that there is neither attenuation nor amplification. The amplification coefficient drops quickly when the vibration frequency gets larger than the eigenfrequency. In Fig. 2d, we demonstrate the amplification coefficient under different loaded mass and input vibration amplitude. When the loaded mass varies, the eigenfrequency as well as the corresponding peak amplification coef-ficient achieved at that frequency changes. As shown in Fig. 2d, as the loaded mass changes from 1.2 g to 32 g, the eigenfrequency varies from 25 Hz to 9.5 Hz. The maximum amplification coefficient of more than 12 is observed at a mass load of 3.5 g. For a fixed mass load, when the input vibration amplitude increases from zero, the maximum ampli-fication near the eigenfrequency first increases and then decreases, showing the nonlinearity of the device. Besides that, the device is tested when attached to a vibrating surface with different inclination angles to the ground, as illustrated in Fig. 2e. With the same input excitation signal, the measured displacement of the top membrane layer is similar for different inclination angles because the influence of the gravita-tional force is small and did not break the resonance state of the system, showing that the device is robust under inclination. Besides the experiments (shown in Supplementary Videos S1 and S2), we have performed numerical simulation to verify that the change in the dis-placement of the membrane is small and negligible under different inclination angles (Supplementary Note 2 and Supplementary Fig. S2 in the Supplementary Materials).

### Robustness of the device
The cycle life of the membrane depends on the membrane material prop-erties as well as environmental factors and mechanical conditions[35]. The membrane used in our device is the super-stretchable abrasion-resistant natural rubber film from McMaster-Carr (Super-Stretchable Abrasion-Resistant Natural Rubber, 8611K1159) with a tensile strength of 27.58 MPa and an ultimate elongation of 750%. Since the device is designed for enhancing the sensitivity of small vibrations, the membrane is strong enough to endure typical deformations in application, as shown in the simulation of the stress distribution on the membrane in the Supplementary Materials (Supplementary Note 3 and Supplementary Fig. S3). Moreover, the amplification factor is not a linear function of the input vibration amplitude. Once the input vibration amplitude is higher than a specific value, the amplification factor also drops (Fig. 2d), setting a safe limit on membrane deformation.

Although the ultimate elongation is beyond reach, the repetitive deformation at the resonance state might shorten the life cycle of the membrane. We conducted a life cycle testing for the device with the reso-nance frequency of 9.5 Hz. The input vibration is at 9.5 Hz. The device worked at its maximum amplification. The top membrane vibration amplitude is measured as 5 mm with a 0.7 mm input vibration amplitude. After letting the membrane vibrate for 24 h ($8 \times 10^5$ cycles), we didn't observe any damage to the membrane or any change in the output vibration amplitude. When the membrane vibrates at higher frequencies such as at 110 Hz, the overall vibration amplitude will be smaller, but the strain rate is

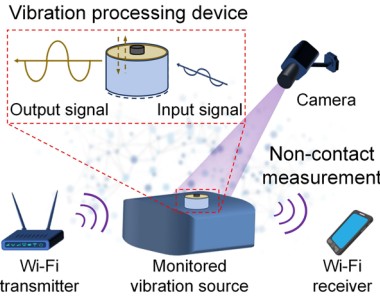

**Fig. 1 | Illustration of the passive vibration processing device and its applications.** The device can be mounted on the surfaces of vibrating subjects to process the vibration signals passively in non-contact vibration sensing such as Wi-Fi or camera based systems.

higher. The impact of vibration frequency and other factors on the life cycle of different membrane materials is a meaningful topic for future research.

The properties related to our device, including density, Young's modulus, Poisson's ratio, tensile strength, and elongation, have negligible changes over the typical environmental conditions, such as temperature[36] in the range of 0–50 °C and humidity[37] in the range of 30% RH to 50% RH, that will be encountered in most of the application scenarios such as healthcare, architecture health diagnosis, and manufacturing process monitoring. However, under some extreme conditions, the rubber film will lose its functions. For example, the film will have aggravated performance decline outside the range of −60 °C to 100 °C. The water molecules in the environment will accelerate the aging of rubber film, making the device not suitable to work in high humidity.

## Analog vibration signal filtering effect of the proposed device

The device can also be considered as an analog vibration signal filter. When the input vibration waveform is more complex than a single-frequency sinusoidal wave, such as the waveforms of human breaths, heartbeats, and other more realistic vibrations, part of the frequencies will be amplified, and part will be diminished. Figure 2f shows a triangle-like input non-sinusoidal vibration waveform with a fundamental frequency of 1 Hz and a peak-to-peak amplitude of 5 mm, provided by the shaker and measured by the laser vibrometer. Its frequency spectrum (black curve in Fig. 2g) contains multiple peaks centered at 1 Hz, 2 Hz, 3 Hz, etc. with decreasing amplitude. Upon filtering by the device with an eigenfrequency of 9.5 Hz, the peak near the eigenfrequency of the device is largely amplified. We observed the change of the frequency peak around the device's eigenfrequency for several different input waveforms with different fundamental frequencies either smaller or larger than 9.5 Hz (Supplementary Fig. S4). This newly adapted frequency peak can be exploited as a label for the device. When there are multiple vibration sources in the sensing environment, for example in remote heartbeat/breath sensing when there are multiple persons in the same room, it was difficult to separate the signals from different persons in existing Wi-Fi remote sensing technologies[38]. We can attach devices with different eigenfrequencies to different vibration sources and use the eigenfrequency labels to differentiate the signals from each device. Similarly, in an industrial setting, the eigenfrequency of the passive device could be used to distinguish one machine from another. In addition, the device opens the possibility of analog vibration filtering and processing that have the potential to reduce digital processing burden and power consumption, improve signal to noise ratio, and enhance system security[39,40].

## Improved sensitivity and labeling effect in wireless Wi-Fi vibration sensing

Wi-Fi is one of the most common wireless communication technologies to transfer data between two devices. In the recent past, it is being considered as a potential tool for passive sensing of the surrounding environment due to the ubiquitous presence of Wi-Fi devices. Particularly, it has been explored

for its potential in contactless vibration sensing[26]. Wi-Fi based vibration sensing utilizes a Wi-Fi router emitting a signal, which is reflected by a vibration source and captured at the receiver, for example, a cellphone or a Wi-Fi router. The receiver then calculates the Channel State Information (CSI), which captures the frequency response of the wireless channel between the Wi-Fi transmitter and the receiver. An accurate estimate of the CSI has been shown to be highly sensitive to changes in the environment[41]. Therefore, CSI is used to calculate the vibration frequency of the surface of interest that is placed between the Wi-Fi transmitter and receiver[42]. However, current Wi-Fi vibration sensing methods are not widely used in practice due to their limited sensitivity. First, it lacks sensitivity to small vibration amplitudes because of the limitation posed by the large wavelengths of Wi-Fi signals. Second, it is unable to differentiate between different objects because widely used Wi-Fi antennas in routers and cellphones lack the needed directivity. In our proposed system, our passive vibration processing device aims to overcome the sensitivity limitation of Wi-Fi based sensing by amplifying the vibration signal and in turn its impact on CSI measurement at the Wi-Fi receiver. The ability of our proposed device to assign a unique eigenfrequency to identify an object in an environment addresses the second limitation; thus, we leverage the benefits of low-cost, non-contact Wi-Fi based sensing, as well as overcome its limitations using our design of the device.

Consider the device with a 9.5 Hz eigenfrequency attached to the surface of a vibrating sample. It amplifies the movement (vibration) amplitude of the top membrane compared to the input vibration of the bottom layer, resulting in a more apparent change in the reflected Wi-Fi signal and its estimated CSI, which is then used by the Wi-Fi receiver to accurately determine the vibrating frequency (shown in Fig. 3a). The methodology for Wi-Fi sensing used in this paper is detailed in the methods section. The frequency of Wi-Fi operation used in this study is 5 GHz with a channel bandwidth of 20 MHz, containing 52 Wi-Fi channels for measurement. In Fig. 3b, we plot the CSI phase difference between two antennas at the Wi-Fi receiver, with and without the amplification device for a single Wi-Fi channel. The phase difference of CSI is a function of the displacement of the vibrating surface. The Fourier frequency spectrum of the CSI phase difference indicates the vibration frequencies and their strength. The variations of CSI phase difference peak-to-peak value, measured over six observations of 9.5 Hz vibrations with vibrating amplitude ranging from 0.1 mm to 6 mm is plotted in Fig. 3c. In the presence of our proposed device, the peak-to-peak amplitude of CSI phase difference is improved even at small vibration amplitude of 0.1 mm. An instance of the frequency spectrum obtained from the CSI phase difference with an input vibration amplitude of 1 mm is shown in Fig. 3d. In the absence of a device attached to the vibration source, the amplitude of the 9.5 Hz peak in the vibration spectrum is lower than that of frequencies corresponding to environmental noise; this low amplitude is below the measurement limit of our Wi-Fi system, resulting in poor performance in detecting vibrations. Upon attachment of the device, the 9.5 Hz peak amplitude is amplified, and the Wi-Fi system accurately measures the real vibration frequency. The accuracy for different vibration amplitudes (measured by the correct rate of Wi-Fi sensing dominant frequency results among different time periods) is also improved with the device, as shown in Supplementary Fig. S5. With the device, vibration amplitudes as low as 1 mm are detected with 100% accuracy, due to its amplification. In contrast, without the device, even vibration amplitudes of 4 mm go unnoticed by the Wi-Fi receiver. Furthermore, we demonstrate the labeling function of the device in Wi-Fi vibration sensing. The signal shown in Fig. 2f is used as the input and the measured vibration frequency spectrum is shown in Fig. 3e. Apart from the input frequency 1 Hz, a 9 Hz labeling frequency peak is also observed, which can be used to label and distinguish the vibration surface.

## Passive sensitivity enhancement of camera-based non-contact vibration measurement

As digital cameras become widely accessible, camera-based detection and monitoring techniques are quickly developed. As a second example of

**Fig. 2 | Characterization of the device. a** Device schematics and a photo of fabricated devices with different resonance frequencies (9.5 Hz, 110 Hz and 285 Hz). **b** Input and output vibration waveforms when the input frequency is 9.5 Hz measured from the device with 9.5 Hz eigenfrequency. **c** Amplification coefficients as a function of input frequency for two devices with 9.5 Hz and 110 Hz eigenfrequency respectively. **d** Amplification coefficient as a function of input vibration amplitude for devices with different loaded mass $m$. **e** Output waveforms when the device has different inclination angles with the ground. **f** Output vibration waveform when the input waveform is non-sinusoidal. **g** Frequency spectrums of the waveforms in **f**.

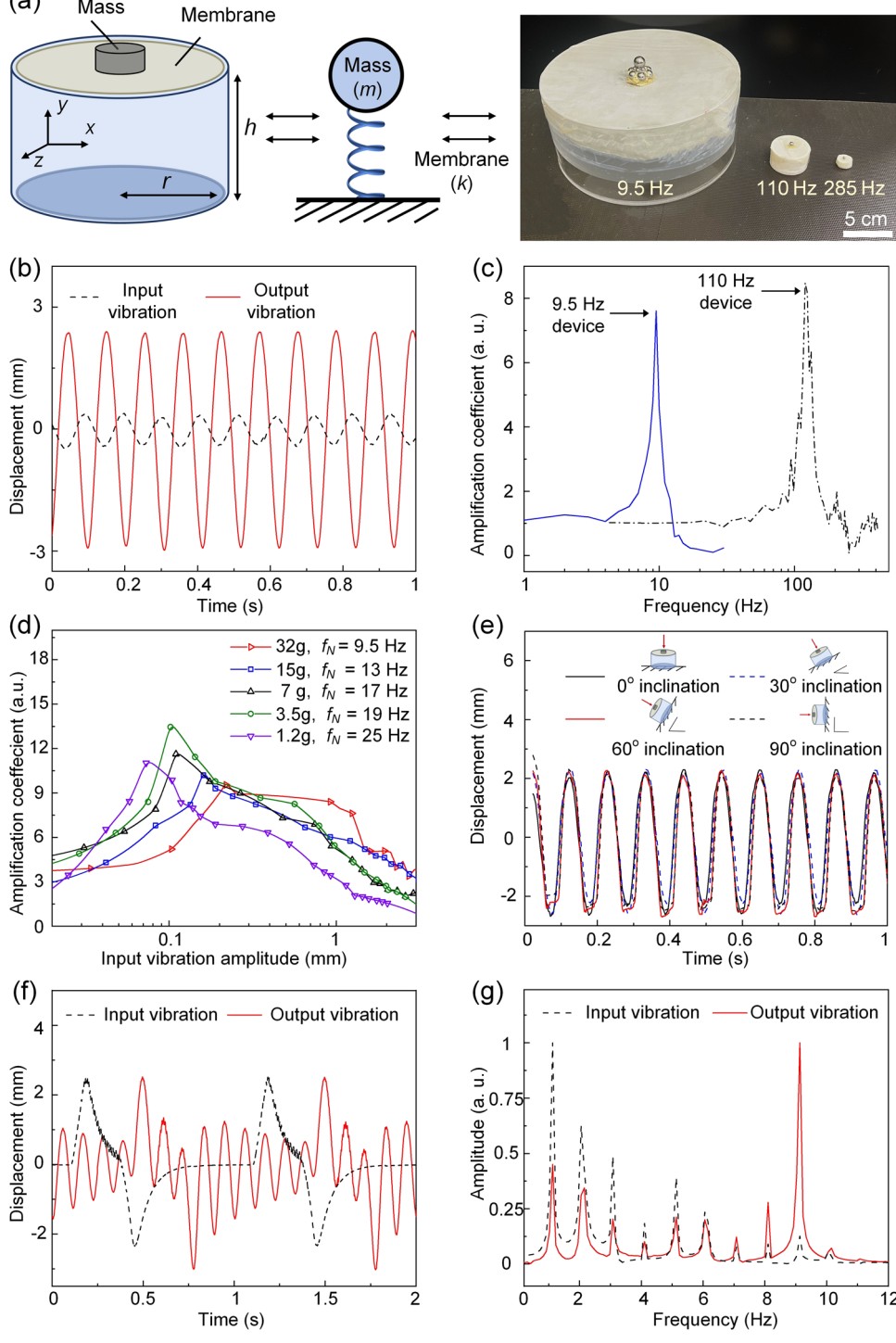

application, we show the device's performance in improving the sensitivity of vibration sensing using a camera (Chronos 2.1-HD). The experiment setup is shown in Fig. 4a. The input vibration frequency and the device eigenfrequency are both set to 110 Hz. The device is mounted on a rigid plate on top of a shaker. To obtain an image with better resolution, a microscope is used in combination with the high-speed camera to record the vibration video of the device. An example video is shown in Supplementary Video S3. The displacement of the sample can be extracted by image processing methods. Here we first make an object segmentation and then analyze the movement of the top membrane layer and bottom layer[43,44] (details are described in Methods). The extracted displacement waveform is plotted in Fig. 4b, where the membrane layer has a more enhanced

movement than the bottom layer. The input and output vibration waveforms are plotted in Fig. 4b. The peak-to-peak amplitudes of input and output vibrations are around 0.1 mm and 0.4 mm, respectively, showing a 4-fold amplification. The frequency spectrums of the input and output vibrations are shown in Fig. 4c. Compared with the bottom layer movement, the top layer has a clear peak at 110 Hz, demonstrating the enhancement of the vibration. The snapshots of the video are shown in Fig. 4d, demonstrating the amplified vibration of the mass on the membrane. Furthermore, we also fabricated another device with a similar structure but having a higher eigenfrequency of 285 Hz. The vibration waveforms are extracted from the recorded video (see Supplementary Videos S4 and S5) by the pixel value changes on the vibration sample boundary and vibration amplitude

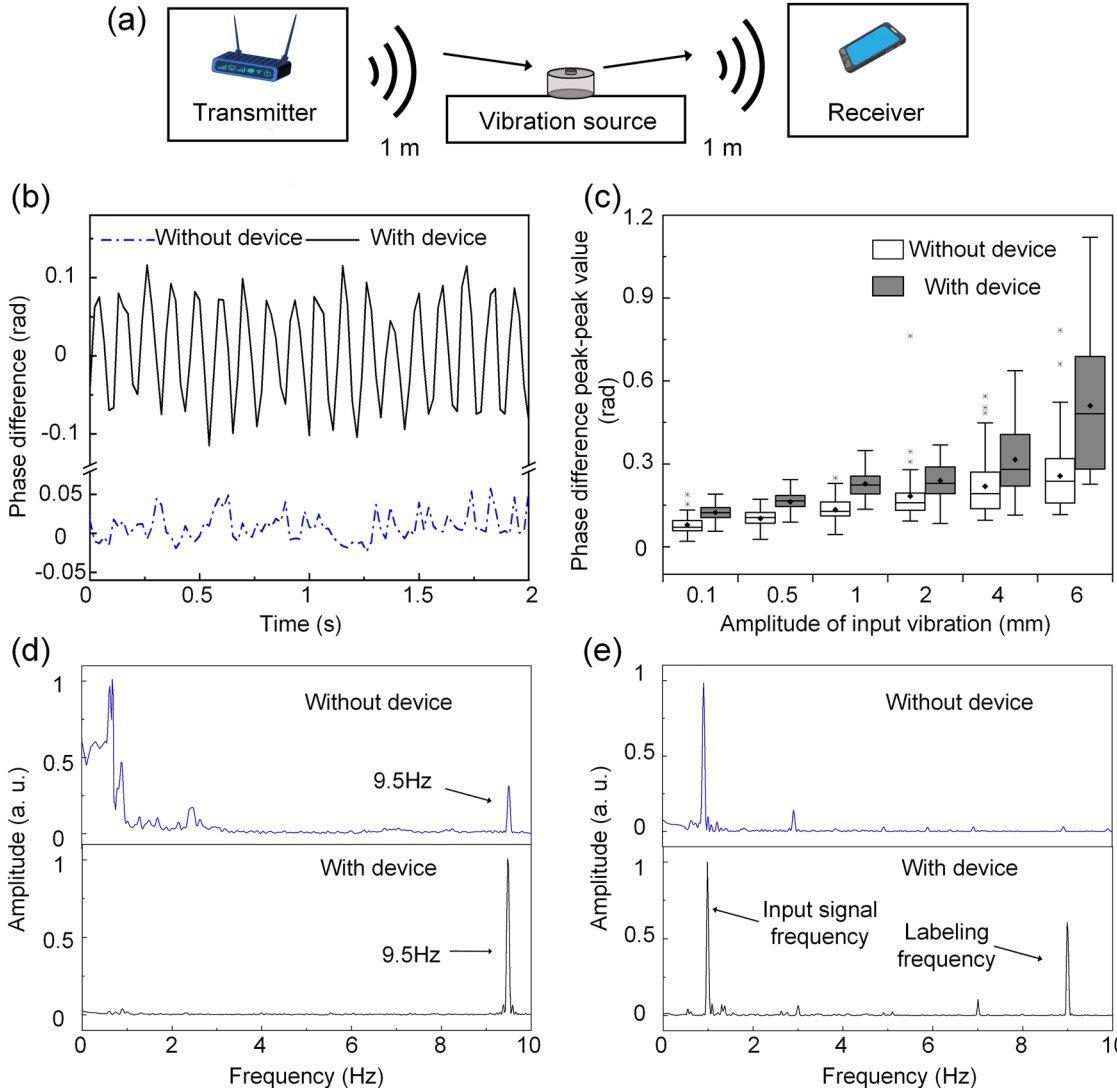

**Fig. 3 | The application of fabricated device in Non-contact Wi-Fi vibration detection. a** A schematic illustration of the device application in Wi-Fi vibration detection, **b** Experimentally measured the Channel State Information (CSI) phase difference between the two receiver antennas listening on the same Wi-Fi channel. The input vibration amplitude is set as 1 mm. **c** Statistic box plot (displays the minimum, first quartile, median, third quartile, and the maximum as a box, mean as a dot in box, and outlier as dots outside the box) of the experimental CSI phase difference peak-peak values for 52 Wi-Fi channels with the 30 s Wi-Fi measurement. The frequency spectrum of the vibration signal measured by Wi-Fi for (**d**) amplification (the input vibration is 9.5 Hz sinusoidal wave with amplitude of 1 mm) and (**e**) labeling (the input signal is non-sinusoidal with amplitude of 5 mm (the same as the one in Fig. 2f).

enhancement of 8.4 folds after applying the device is also observed (details are presented in Supplementary Fig. S7).

**Multifunctional editing of the device**

Apart from the amplification and filtering functions, the device also provides a platform for surface editing that can add more functions to the sensing system. In this paper, we demonstrate two examples of surface editing.

For the Wi-Fi-based vibration sensing, a directional reflector (shown in Fig. 5a) is attached to the top membrane of the device to replace the loaded mass in the original design. The new device has an eigenfrequency of 5 Hz. The reflector is composed of a vertical plate connected to a horizontal plate, which can modify the directivity and the far-field radiation pattern of the reflected Wi-Fi signal[45]. This influence about wave intensity on the direction can be used in Wi-Fi vibration sensing. In the experiment using the reflector, the input vibration frequency is at the eigenfrequency of 5 Hz. The position of Wi-Fi transmitter and receiver are fixed and we rotate the reflector to different orientations with the rotation angle θ. At each orientation, the measured peak-to-peak phase difference between two antennas across all 52

Wi-Fi channels is shown in Fig. 5b. The results have clear angle-dependence. The first and fifth orientations have larger phase differences due to the reflector directivity. Our Wi-Fi signal reflector example provides the possibility of surface modifications to improve wireless reception quality, reduce energy consumption, and achieve better security and privacy.

For the camera vibration sensing, a fluorescent layer is coated on the surface of the loaded mass in the device with eigenfrequency of 125 Hz. The input vibration has a frequency of 125 Hz. The pictures of the coated device in bright and dark environments are shown in Fig. 5c. Because of the fluorescent coating, the vibration of the device can still be captured by a camera in dark environments, as demonstrated in Fig. 5d, e, as well as the recorded video (see Supplementary Videos S6 and S7).

**Tuning of resonance frequency**

The resonance frequency at which the device produces the largest amplification can be adjusted within a broad frequency range by changing the device parameters, including the dimensions and material properties of the membrane, and the weight of the loaded mass. Figure 6 shows the first

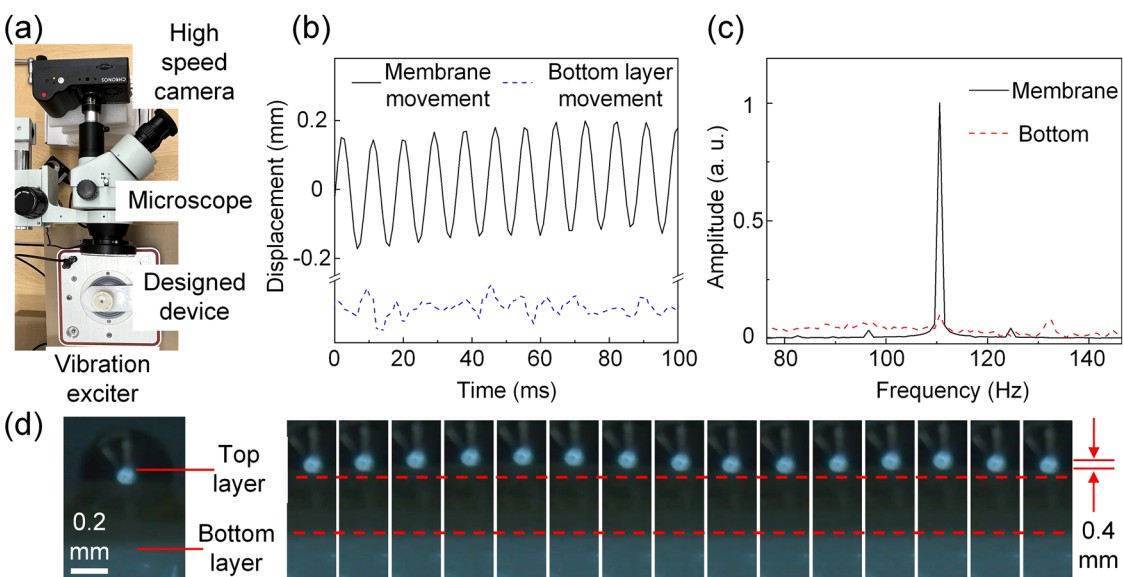

**Fig. 4 | Vibration amplification device for camera-based vision method. a** Experiment setup. **b** The extracted vibration waveforms from Supplementary Video S3 (the eigenfrequency of the device is 110 Hz). **c** Frequency spectrums of the vibration waveforms in **b**. **d** Screenshots of the vibration video at different times (interval = 1 ms).

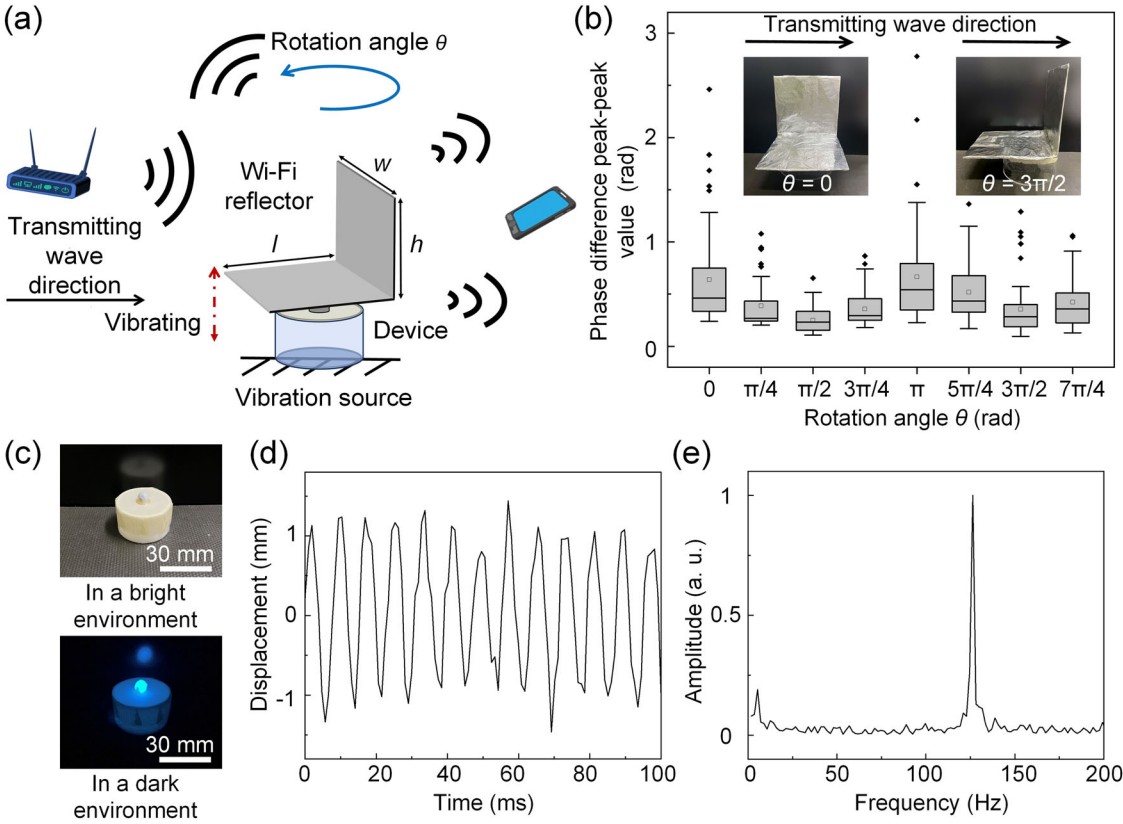

**Fig. 5 | Multifunctional editing of the device: directional Wi-Fi reflector and fluorescent central load. a** Demonstration of the designed Wi-Fi reflector mounted on the top of the device in Wi-Fi sensing and the control of wave by different rotation angle. **b** Box plot of the measured the Channel State Information (CSI) phase difference peak-peak value of 52 Wi-Fi channels with eight different reflector rotation angles with the 30 s experiment. The inserts are the experiment setup demonstration of the experiments with different top layer rotation angle. All box plots show the minimum, first quartile, median, third quartile, and maximum, with the mean as a dot inbox, and outlier as the dots outside the box. **c** the pictures of fluorescent layer coated device in bright environment and dark environment (the corresponding video is Supplementary S6 and S7, respectively). **d** The extracted vibration waveform from Supplementary Video S7 of fluorescent-coated device in dark environment. **e** Frequency spectrum of **d**.

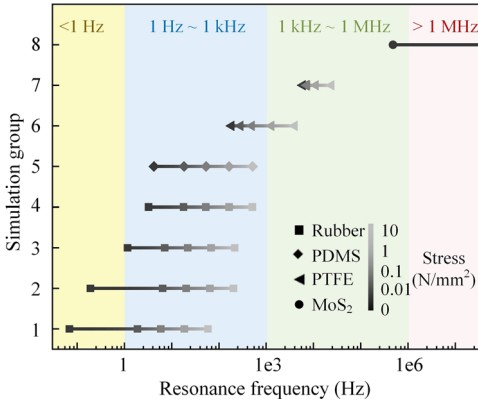

**Fig. 6 | Eigenfrequency tuning results with different parameters of the device.** Eight groups of simulated eigenfrequencies are shown, where in each group different stresses from 0 to 1 N·mm$^{-2}$ are added on the membrane. The details of the device parameters, including Young's modulus, diameter, and thickness of the membrane and weight of the loaded mass are listed in Table 1.

eigenfrequency of the device with eight groups of parameters listed in Table 1. Four different membrane materials that can be used to fabricate the device in different applications are chosen in different simulation groups: Rubber[46], Polydimethylsiloxane (PDMS)[47], polytetrafluoroethylene (PTFE)[48], and MoS$_2$[49]. The diameter ranges from 0.03 mm to 80 mm, the thickness from 0.015 μm to 150 μm, and the weight of the loaded mass from zero (no mass) to 23.285 g. Furthermore, an external stress ranges from zero to 10 N·mm$^{-2}$ is added to the membrane to simulate the tension in the membrane. By tuning one or multiple parameters, the eigenfrequency can span several orders of magnitude, making our device applicable in a broad range of vibration sensing applications, such as human vital sign sensing[25,50,51] and natural disaster monitoring[52] usually below tens of hertz, structure health inspection[18,19,21,22,53–59] in the range of hertz to hundreds of hertz, and material characterization in MHz and even GHz range[49]. In the Supplementary Fig. S8, we provide a systematic study of the eigenfrequency tuning with the above parameters in the range of 0-100 Hz.

## Conclusion

This study presents a metamaterial-based passive analog vibration signal processing device mounted on the vibration surface for real-time, low-cost, and wireless vibration sensing. The device performs analog amplification and labeling on vibration signals, which are demonstrated in Wi-Fi- and camera-based vibration detection. Most of the existing works on non-contact vibration measurement have focused on software or signal processing approaches on the data acquired[24,38]. Our design of a hybrid sensing system, which leverages a passive physical device attached to the vibrating surface, improves the accuracy of the sensed data physically thus reduces the overhead on digital signal processing needs. The proposed device is composed of a mass-loaded membrane, whose vibration eigenfrequency demonstrated in

this work are 9.5 Hz, 110 Hz and 285 Hz. It is relatively straight forward to expand the range of eigenfrequencies to be lower than 1 Hz or to as high as few MHz by modifying the device dimensions and the fabrication methods, in order to meet the working demands of various applications.

Our device can be attached to a vibrating surface using glue, tape, magnets, or other methods, as long as the bonding between the device and the vibrating surface is stable. Furthermore, the vibrating surface does not need to be flat. We can edit the bottom surface of the device to match that of the vibrating source. We demonstrated up to 13.35 folds of vibration amplitude amplification. The amplification coefficient can be further improved by optimizing the combination of membrane material properties, membrane size, and the loaded mass, as well as by choosing a membrane material with lower vibration damping. The direct application of our device is to improve the sensitivity of non-contact vibration sensing systems, such as Wi-Fi/RFID- or camera-based vibration sensing systems. It is well known that high frequency vibrations usually have smaller vibration amplitudes[60], limiting the highest detectable frequency by vibration sensing systems. With the help of our vibration amplitude amplification device, the maximum detectable vibration frequency range can be expanded.

Our device serves as a physical layer filter for the vibration signals. In this paper, we demonstrated that a frequency peak near the first eigenfrequency of the device will always appear in the spectrum of the output vibrations. This frequency peak can be used as a label to address another major challenge in wireless vibration measurement: source identification. It is difficult to distinguish two vibrating sources using state-of-the-art wireless sensing techniques. Our proposed device addresses this problem by assigning a unique eigenfrequency to the device that is attached to each source in an environment. For example, in structure health monitoring applications, multiple devices can be attached to different positions of the structure to sense the position-dependent vibration modes inside the structure remotely. The data can be used to monitor not only the presence of structure damages but their locations. Beyond the labeling function, more filtering functions can be developed in the future by manipulating the frequency response of the device. For example, the higher order eigenmodes that have not been considered in this paper have richer properties to be explored for analog vibration signal processing.

Multifunctional editing of the device's top layer will add more functions to wireless vibration sensing systems. In a Wi-Fi based system, we integrated a directional Wi-Fi reflector to our device to provide direction-dependent sensitivity. This additional reflector can be used to improve wireless reception quality and achieve better security. In a camera-based system, we added fluorescent material to achieve vibration detection in a dark environment. From the material prospect, the membrane material can also be designed sensitive to environmental change, for instance, temperature or density of chemical compound. As a result, the remote monitoring of environment can be realized. The flexibility of our device in terms of its shape, size, and materials render it as a suitable platform for surface editing for multi-functional vibration sensing.

In conclusion, we demonstrated a fully passive, metamaterial-based vibration processing device that can make wireless vibration sensing more

## Table 1 | Details of parameters in eigenfrequency tuning

| Simulation group | Young's modulus: Pa | Diameter (mm) | Thickness (μm) | Weight of the loaded mass (g) | Stress (N·mm$^{-2}$) |
|---|---|---|---|---|---|
| 1 | Rubber ($1.5 \times 10^6$) | 80 | 150 | 23.285 | 0, 0.01, 0.1, 1, 10 |
| 2 | Rubber ($1.5 \times 10^6$) | 80 | 150 | 0.44 | |
| 3 | Rubber ($1.5 \times 10^6$) | 80 | 1500 | 0.44 | |
| 4 | Rubber ($1.5 \times 10^6$) | 30 | 150 | 0.44 | |
| 5 | PDMS ($2.5 \times 10^6$) | 30 | 150 | 0.44 | |
| 6 | PTFE ($5.24 \times 10^8$) | 30 | 150 | 0.44 | |
| 7 | PTFE ($5.24 \times 10^8$) | 3 | 150 | 0 | |
| 8 | MoS$_2$ ($2.5 \times 10^{11}$) | 0.03 | 0.015 | 0 | |

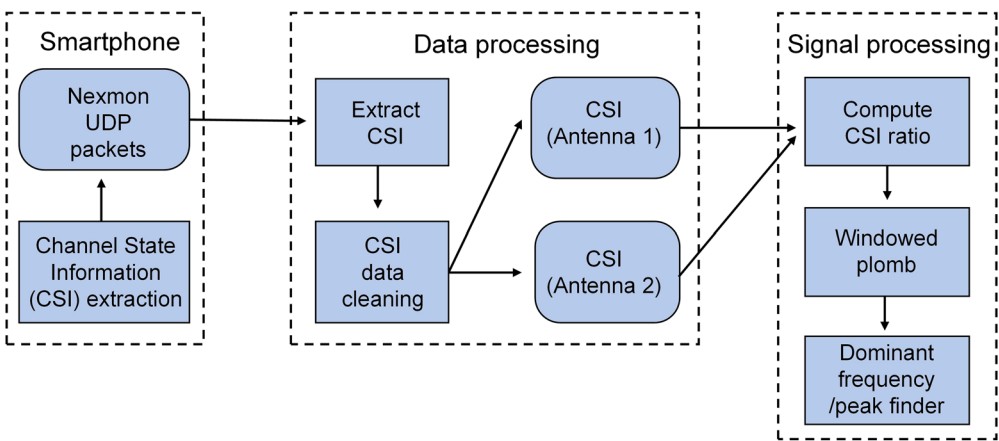

**Fig. 7 | An overview of the signal processing procedures in the Wi-Fi sensing system.** An off-the-shelf smartphone is used to capture and process the Wi-Fi signals to extract the spectra of the vibrations after being reflected by the device.

accessible and benefit a wide range of applications in personalized healthcare, industrial structure inspection, natural disaster monitoring, and material characterization, and promote the integration of sensing capabilities into the rapid development of wireless communication and cloud computing infrastructures.

## Methods

### Device fabrication and laser vibrometer measurement setup

The membrane of the device is made of thin natural rubber film (McMaster-Carr, Super-Stretchable Abrasion-Resistant Natural Rubber, 8611K11) with a thickness of 0.1524 mm. The Young's modulus of the film within 100% elongation is 0.83 MPa and the tear strength is 1.79 MPa. The film can be elongated 750% at break with an ultimate tensile strength of 27.58 MPa. The masses load at the center of the membrane are steel balls of different sizes. The balls can be stuck together to reach a proper weight as designed. The cylindrical hard shell of the device with 9.5 Hz eigenfrequency is composed of acrylic plastic components and the shells of other smaller devices are fabricated by 3D printing. The mass-loaded membrane is cut to the same circular shape as the cylindrical shell and mounted on the top of the shell. The detailed parameters of the designed devices and their corresponding eigenfrequencies can be found in Supplementary Table S1. After cutting to the same circular shape with hard shell, the rubber film is stuck on the top of shell with the mass stuck in the center. The directional reflector is made of hard cardboard covered with tinfoil. For the fluorescent coating layer, fluorescence particles (SEISSO company) are added to the silicone rubber solution (PIXISS company) and coated on the surface of the steel ball that is used as the loaded mass of the device.

The experimental setup for characterizing the devices' vibration properties can be viewed in Supplementary Fig. S1. The input vibration waveform is generated from a function generator (Tektronix AFG3022C) and then input to a power amplifier. The signal from the power amplifier controls a shaker (Bruel & Kjaer Type 4809) to provide the input vibration to the hard plate, where the device is attached. A laser vibrometer system (Polytec OFV 056 with scanning head PSV-200) is used to measure the transverse vibration motion of the mass-loaded membrane when device is attached, and the supporting plate motions without the device.

### Wi-Fi measurement setup

The signal processing procedures in the Wi-Fi sensing system are illustrated in Fig. 7. The CSI in Wi-Fi sensing is mathematically represented as the superposition of a signal from all the N paths as shown in Eq. (1)

$$H(f;t) = \sum_n^N a_n(t) e^{-j2\pi f \tau_n(t)} \qquad (1)$$

where $a_n(t)$ represents the amplitude attenuation factor, $\tau_n(t)$ is the phase of the electromagnetic signal due to the propagation delay (imaginary component of a particular path), and $f$ is the carrier frequency. The CSI amplitude ($|H|$) and phase ($\angle H$) are impacted by the displacements and movements of the transmitter, receiver, and surrounding objects and humans. Therefore, CSI effectively represents the medium of signal propagation and any changes to CSI in a stable environment resulting from movement of an individual or the device in this experiment can be detected. The device movement results in periodic changes to the phase and should therefore be the dominant frequency in the frequency response of the phase difference signal measured across time.

In this study, the commercial Asus RT-AC86U Wi-Fi router is used as the transmitter and captured by a rooted smartphone (Nexus 6 P) with bcm4358 Wi-Fi chip. The Wi-Fi beacon interval is set to 20 ms and distances between router and measured sample, sample and smartphone are both set as 1 m. The measured receiving signal strength is −25 to −30 dB and the time duration of one vibration measurement is set as 30 s. As Wi-Fi 4 or 802.11n utilizes OFDM (Orthogonal Frequency Division Multiplexing) for transmission, 52 sub-carrier (or channels) CSI across 20 MHz can be computed. The Nexmon CSI tool is used to extract the CSI at the receiver for each antenna. We perform basic preprocessing such as angle unwrap and time-based correlation between the packets received on each antenna at the receiver[41].

CSI phase difference between two antennas at the receiver is then calculated and used to determine the amplitude and frequency of vibration of the surface of interest. As the signal is measured irregularly, we apply plomb, a MATLAB based implementation of the Lomb–Scargle Periodogram[61,62], to obtain the Power Spectral Density (PSD), which is further analyzed to obtain the dominant frequency in a given frequency range. The whole procedure of the dominant frequency calculation can be viewed in Supplementary Algorithm Table S2. The computations are performed using a sliding window protocol to limit the time frame, over which the plomb is applied to improve its accuracy. The time window is set as 15 s and the window slides forward by 20 ms. The accuracy results shown in Fig. 3 is averaged over a large number of time windows getting the right vibration frequency and the total time window number is used to calculate accuracy.

### Video recording and vibration extraction

The video recording system for camera-based vibration sensing is shown in Fig. 4a. To achieve better video resolution, a microscope with 5 times amplifications is used when recording Supplementary Video S3 while 20 times amplifications for Supplementary Videos S4 and S5. Supplementary Videos S6 and S7 are recorded without the microscope. The camera (Chronos 2.1-HD) for recording the videos has frames per

second (fps) of 1069.61. The camera lens is set to be parallel to the vibrating surface. Two video processing methods are used to extract the vibration waveform from the recorded videos. The first one is based on vibrating object segmentation and moving trajectory extraction. As shown in the Supplementary Fig. S6a–c, a snapshot is first converted to grayscale picture with value 0–255 to represent the color in every pixel. The moving sample is then segmented by a threshold and the centroid of it can be calculated. Movement of this centroid with time (shown in Supplementary Fig. S6d) can be used to analyze the vibration frequency. This method is used for the 110 Hz device. Furthermore, the grayscale value changes on the vibration sample boundary are used to acquire vibration frequency for the 285 Hz device, which can deal with the smaller vibration movement compared to the first method[63]. The edge of the moving sample (where the grayscale value has the largest variation) is first obtained for both top layer central mass and bottom hard shell. Points on these edges are chosen and the color value changes with time at these points are plotted in Supplementary Fig. S7d, e.

## Data availability
All data needed to evaluate the conclusions in the paper are present in the paper and/or the Supplementary Materials. Additional data related to this paper may be requested from the corresponding author.

## Code availability
The full version of the code may be available from the corresponding author upon reasonable request.

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

## Acknowledgements

C.M. and D.Z. acknowledge the funding support from Wisconsin Alumni Research Foundation and NSF through the University of Wisconsin Materials Research Science and Engineering Center (DMR-1720415). B.K., A.P., and M.O.S. acknowledge the funding support from the following NSF grants: CCSS-2034415, CNS-2107060, CNS-2142978, CNS-2213688, CNS-2112562, and Wisconsin Alumni Research Foundation awards. The authors would like to acknowledge Ji Zhou, Weikang Xian, and Prof. Ying Li from Department of Mechanical Engineering of the University of Wisconsin-Madison for their valuable discussion.

## Author contributions

D.Z., C.M., and B.K. conceived the research idea. D.Z. performed the mechanical simulation and fabricated the devices. D.Z., A.P., and M.O.S. performed the experiments, collected the data, and did the analysis. D.Z., C.M. A.P. and B.K. contributed to the paper writing. C.M. supervised the study.

## Competing interests

The authors declare no competing interests.
