## [Peer Review File · Communications Engineering]

Reviewers' comments:

Reviewer #1 (Remarks to the Author):

This paper introduces a passive metamaterial-based analog processor to improve the sensitivity of wireless vibration sensing techniques. The proposed device can also be considered an analog vibration signal filter. The paper holds scientific relevance and is of interest to the journal's readers. However, there are some comments the authors are encouraged to address to complement their work:

1-For the next version of your manuscript, please add line numbers to your manuscript.

2-In the Abstract:

---instead of "at designated frequencies," please clearly mention the specific frequency range that you studied in your paper.

---provide examples for the statement "broadening its applicability to different applications."

3-Reaching a higher amplification factor (e.g.,10) may be achievable, but it's not practical for the long-term application due to membrane's cycle life limitations. The authors are encouraged to share insights from their experiments about the effects of the frequency range on the membrane's cycle life for their test samples.

4-On page 3, "frequency identification (RFID)" needs to be changed to "radio frequency identification (RFID)"

5-The attached mass enriches the vibration amplification factor, but it also causes stress concentration on top of the membrane and accelerates the tensile/bending failure of the membrane. So, the authors should clarify that the maximum stress across the membrane doesn't exceed its tensile strength in their tests. This can be accomplished through either experimental data or a basic finite element analysis.

6-This paper needs to include a comprehensive Parametric Study to establish a connection between the device's range of eigenfrequencies, its dimensions as well as material properties. As a reference for future design, the results of this study can then provide specific insights into which range of design/physical parameters can be used to tune the membrane's frequency.

7-Reduce the introduction to a maximum of 2 pages at most.

8-No need to mention "more details in design and characterization of the vibration amplification device" that you repeated on pages 5 and 6. Readers typically check for design novelty in the Design section.

9-I highly recommend that the authors add a Conclusion section to their manuscript.

Reviewer #2 (Remarks to the Author):

This paper is generally well-written and addresses interesting points in wireless vibration sensing. I have some comments and observations which need to be addressed before further review:

- My main concern is about the metamaterial the authors use for the membrane of the sensor.
 - o What is the metamaterial of the membrane that the sensor is made of?
 - o How is this metamaterial manufactured?
 - o Why are the authors calling this a metamaterial?
 - o Is this metamaterial 3d printed and what are the limitations of it? What is the longevity of this material to fatigue or excess deformation?

o How repetitive is the process of manufacturing the metamaterial?

- The authors need to provide more information here and experimental testing results on the mechanical properties of this material.
- Is this metamaterial sensitive to the environment?
- It is surprising that the sensor works in all angles, as the stiffness of the metamaterial is far from rigid, the reviewer is guessing. I understand that the sensor can work in 0 and 180 degrees, but not for others. The authors need to elaborate more into this and explain why this is happening.
- The reviewer does not see why this sensor would be used for Wi-Fi identification. Aren't there much cheaper and easier ways to identify Wi-Fi? Why is this sensor important for that task?
- The reviewer would like to see more applications on structural health monitoring and how could this technology be applied on an actual structure, given the connectivity to a surface challenge. Also, the authors mention that these sensors could work collaboratively to provide more info. Can the authors be more specific on this topic? Examples would be preferred here.

Replies to Reviewer 1:

This paper introduces a passive metamaterial-based analog processor to improve the sensitivity of wireless vibration sensing techniques. The proposed device can also be considered an analog vibration signal filter. The paper holds scientific relevance and is of interest to the journal's readers. However, there are some comments the authors are encouraged to address to complement their work:

Thank you for the valuable comments and suggestions on our manuscript. We have addressed your comments and incorporated suggestions in the paper. Please find our response and the corresponding changes to the paper detailed below.

Comment 1:

For the next version of your manuscript, please add line numbers to your manuscript.

Answer: We have added line numbers in the revised manuscript.

Comment 2:

In the Abstract:

----instead of "at designated frequencies," please clearly mention the specific frequency range that you studied in your paper.

----provide examples for the statement "broadening its applicability to different applications."

Answer: We have revised the abstract to add the following sentences:

"In this work, we proposed a fully passive, metamaterial-based vibration processing device, fabricated three prototypes working at 9.5 Hz, 110 Hz and 285 Hz, and verified that the device can improve the sensitivity of wireless vibration measurement methods by more than ten times when attached to vibrating surfaces."

"Finally, the working frequency of the device is widely adjustable over orders of magnitudes, broadening its applicability to different applications, such as structural health diagnosis, disaster warning, and vital signal monitoring."

Comment 3:

Reaching a higher amplification factor (e.g., 10) may be achievable, but it's not practical for the long-term application due to membrane's cycle life limitations. The authors are encouraged to share insights from their experiments about the effects of the frequency range on the membrane's cycle life for their test samples.

Answer: We agree that the cycle life is an important factor toward application. The cycle life of the membrane depends on the membrane material properties as well as environmental factors and mechanical conditions^[1]. The membrane used in our device is the super-stretchable abrasion-resistant natural rubber film from McMaster-Carr (Super-Stretchable Abrasion-Resistant Natural Rubber, 8611K1159) with a tensile strength of 27.58 MPa and an ultimate elongation of 750%. Since the device is designed for enhancing the sensitivity of small vibrations, the membrane is strong enough to endure typical deformations in application (please see details in our response to Comment 5). Moreover, the amplification factor is not a linear function of the input vibration amplitude. Once the input vibration amplitude is higher than a specific value, the amplification factor drops significantly (Fig. 2d), setting a safe limit on membrane deformation.

Although the ultimate elongation is beyond reach, the repetitive deformation at the resonance state might shorten the life cycle of the membrane. We conducted a life cycle testing for the device with the resonance frequency of 9.5 Hz. The input vibration is at 9.5 Hz. The device worked at its maximum amplification. The top membrane vibration amplitude is measured as 5 mm with a 0.7 mm input vibration amplitude. After letting the membrane vibrate for 24 hours (8×10^5 cycles), we didn't observe any damage to the membrane or any change in the output vibration amplitude.

When the membrane vibrates at higher frequencies such as at 110 Hz, the overall vibration amplitude will be smaller, but the strain rate is higher. The impact of vibration frequency and other factors on the life cycle of different membrane materials is a meaningful topic. We expect future research on it.

Fig. R1. Photo of the device at the beginning and end of the life cycle testing.

We have added the above discussion in the section “Robustness of the device” of the revised manuscript on page 9.

[1] Qiu, X., Yin, H., Xing Q. Research progress on fatigue life of rubber material. *Polymers*, 14, 4592 (2022).

Comment 4:

On page 3, “frequency identification (RFID)” needs to be changed to “radio frequency identification (RFID)”

Answer: We have corrected the error in the revised manuscript.

Comment 5:

The attached mass enriches the vibration amplification factor, but it also causes stress concentration on top of the membrane and accelerates the tensile/bending failure of the membrane. So, the authors should clarify that the maximum stress across the membrane doesn't exceed its tensile strength in their tests. This can be accomplished through either experimental data or a basic finite element analysis.

Answer: We have added more information about the membrane in the device fabrication section. The tensile strength of film is 27.58 MPa while ultimate elongation can be 750%. A numerical simulation is performed to check the stress on the membrane. We use the Mooney-Rivlin model for the rubber thin film simulation, where the parameters $C10$ and $C01$ can be calculated approximately by these two equations:

$$E=8\times(C10+C01)^{[2]}$$

$$C01=0.2\times C10^{[3]}$$

where Young's Modulus E can be obtained from material technical data sheet as 0.83 MPa. In the simulation conducted by COMSOL Multiphysics 6.0 structural mechanics module, $C10$ is set as 0.086 MPa and $C01$ is set as 0.017 MPa. The diameter of the circular membrane is 200 mm and the thickness is 150 μm , the same as the membrane used in the fabricated device. We simulated two cases, one with a point load at the center, and the other with a 3 mm-diameter circular load. In both cases, the displacement at the center is 9 mm, which is the maximum displacement of the membrane with 9.5 Hz eigenfrequency reached in our experiment (shown in Fig. 2(d)). The stress distribution on the membrane is shown in Fig. R2. The peak stress on the membrane is 1.98 MPa at the center for the point-load case, and 0.037 MPa for the circular-load case, which are both much smaller than the tensile strength of the rubber film.

Fig. R2. Simulated stress distribution when the maximum membrane displacement is 9 mm, with a point load (a), and 3 mm-diameter circular load (b) at the center of the membrane. The peak stress on the membrane is 1.98 MPa for the point-load case, and 0.037 MPa for the circular-load case.

The following content about the membrane material is added in the “Device Fabrication and laser vibrometer measurement setup” section of the revised manuscript on page 23:

“The membrane of the device is made of a thin natural rubber film (McMaster-Carr Supply Company, Super-Stretchable Abrasion-Resistant Natural Rubber, 8611K11⁶¹) with a thickness of 0.1524 mm. The Young’s modulus of the film within 100% elongation is 0.83 MPa and the tear strength is 1.79 MPa. The film can be elongated 750% at break with an ultimate tensile strength of 27.58 MPa.”

The simulation and discussion about the largest stress in the membrane is added to the revised Supplementary Materials.

[2] Bergstrom, J. S. Mechanics of solid polymers: theory and computational modeling (Chapter 5). William Andrew, 2015.

[3] Krmela1, J., Artyukhov, A., Krmelová, V., Pozovnyi, O. Determination of material parameters of rubber and composites for computational modeling based on experiment data. Journal of Physics: Conference Series, 1741, 012047 (2021).

Comment 6:

This paper needs to include a comprehensive Parametric Study to establish a connection between the device's range of eigenfrequencies, its dimensions as well as material properties. As a reference for future design, the results of this study can then provide specific insights into which range of design/physical parameters can be used to tune the membrane's frequency.

Answer: Thanks for the valuable suggestion. This will indeed be useful to include in the paper. We have revised the manuscript and added a new section at page 19 named “Tuning of the Eigenfrequency”. In this section, we added more simulation results to demonstrate the relations between the device's range of eigenfrequencies and its dimensions as well as material properties.

Fig. 6. Eigenfrequency tuning results with different parameters of the device. Eight groups of simulated eigenfrequencies are shown, where in each group different stresses from 0 to 1 N/mm² are added on the membrane. The details of the device parameters, including Young’s modulus, diameter, and thickness of the membrane and weight of the loaded mass are listed in Table 1.

Table 1. Details of parameters in eigenfrequency tuning.

Simulation group	Young's modulus: Pa	Diameter (mm)	Thickness (μm)	Weight of the loaded mass (g)	Stress (N/mm²)
1	Rubber (1.5×10^6)	80	150	23.285	0, 0.01, 0.1, 1, 10
2	Rubber (1.5×10^6)	80	150	0.44	
3	Rubber (1.5×10^6)	80	1500	0.44	
4	Rubber (1.5×10^6)	30	150	0.44	
5	PDMS (2.5×10^6)	30	150	0.44	
6	PTFE (5.24×10^8)	30	150	0.44	
7	PTFE (5.24×10^8)	3	150	0	
8	MoS ₂ (2.5×10^{11})	0.03	0.015	0	

“The resonance frequency at which the device produces the largest amplification can be adjusted within a broad frequency range by changing the device parameters, including the dimensions and material properties of the membrane, and the weight of the loaded mass. Fig. 6 shows the first eigenfrequency of the device with eight groups of parameters listed in Table 1. Four different membrane materials that can be used to fabricate the device in different applications are chosen in different simulation groups: Rubber⁴⁶, Polydimethylsiloxane (PDMS)⁴⁷, polytetrafluoroethylene (PTFE)⁴⁸, and MoS₂⁴⁹. The diameter ranges from 0.03 mm to 80 mm, the thickness from 0.015 μm to 150 μm, and the weight of the loaded mass from zero (no mass) to 23.285 g. Furthermore, an external stress ranges from zero to 10 N/mm² is added to the membrane to simulate the tension in the membrane. By tuning one or multiple parameters, the eigenfrequency can span several orders of magnitude, making our device applicable in a broad range of vibration sensing applications, such as human vital sign sensing^{25,50,51} and natural disaster monitoring⁵² usually below tens of hertz, structure health inspection^{18,19,21,22,53–59} in the range of hertz to hundreds of hertz, and material characterization in MHz and even GHz range⁴⁹. In the Supplementary Fig. S8, we provide a systematic study of the eigenfrequency tuning with the above parameters in the range of 0-100 Hz.”

Comment 7:

Reduce the introduction to a maximum of 2 pages at most.

Answer: Thanks for your advice. We have reduced the length of the introduction.

Comment 8:

No need to mention "more details in design and characterization of the vibration amplification device" that you repeated on pages 5 and 6. Readers typically check for design novelty in the Design section.

Answer: Thanks for your advice. We have modified the paper.

Comment 9:

I highly recommend that the authors add a Conclusion section to their manuscript.

Answer: Thanks for your advice. We added a conclusion section to the manuscript.

Replies to Reviewer 2:

This paper is generally well-written and addresses interesting points in wireless vibration sensing. I have some comments and observations which need to be addressed before further review:

Thank you for the valuable comments and suggestions on our manuscript. We have addressed your comments and incorporated suggestions in the paper. Please find our response and the corresponding changes to the paper detailed below.

Comment 1:

- *My main concern is about the metamaterial the authors use for the membrane of the sensor.*
 - o What is the metamaterial of the membrane that the sensor is made of?*
 - o How is this metamaterial manufactured?*
 - o Why are the authors calling this a metamaterial?*
 - o Is this metamaterial 3d printed and what are the limitations of it? What is the longevity of this material to fatigue or excess deformation?*
 - o How repetitive is the process of manufacturing the metamaterial?*

Answer: In our manuscript, the word “metamaterial” refers to the mass-loaded membrane (i.e., the circular membrane with a mass attached at the center). In general, metamaterials are artificial structures that are much smaller compared to the wavelength of the wave it interacts with. A mass-loaded membrane is widely used in metamaterials to produce a resonance frequency at which the wavelength is much larger than the membrane size^[4]. Such a deep-subwavelength resonance endows the metamaterial interesting properties such as negative wave refraction^[5] and

enhanced wave absorption^[6]. In our case, the mass-loaded membrane produces the vibration amplification and filtering functions as well as gives our device a compact size.

We have revised the device fabrication section and added more details. The hard shell of the 9.5 Hz-eigenfrequency device is made of an acrylic plastic tube and that of the smaller devices are fabricated by 3D-printing. The membranes used in all the devices are made of commercial rubber film with a tensile strength of 27.58 MPa and an elongation limit of 750%. We conducted a life cycle testing for the device with the resonance frequency of 9.5 Hz. The input vibration is at 9.5 Hz. The device worked at its maximum amplification. The top membrane vibration amplitude is measured as 5 mm with a 0.7 mm input vibration amplitude. After letting the membrane vibrate for 24 hours (8×10^5 cycles), we didn't observe any damage to the membrane or any change in the output vibration amplitude.

We revised the description of metamaterials and mass-loaded membrane in “Working principle of the hybrid monitoring system” section on page 5:

“A mass-loaded membrane is widely used in metamaterials to produce a resonance frequency at which the wavelength is much larger than the membrane size³⁰. Such a deep-subwavelength resonance endows the metamaterial interesting properties such as negative wave refraction³³ and enhanced wave absorption³⁴. In our case, the mass-loaded membrane produces the vibration amplification and filtering functions as well as gives our device a compact size.”

The following content about device fabrication is revised in page 23:

“The membrane of the device is made of a thin natural rubber film (McMaster-Carr Supply Company, Super-Stretchable Abrasion-Resistant Natural Rubber, 8611K11⁶¹) with a thickness of 0.1524 mm. The Young's modulus of the film within 100% elongation is 0.83 MPa and the tear strength is 1.79 MPa. The film can be elongated 750% at break with an ultimate tensile strength of 27.58 MPa. The mass attached at the center of the membrane is composed of steel balls glued together to reach a designed weight. The cylindrical shell of the device with 9.5 Hz eigenfrequency is an acrylic plastic tube and that of other smaller devices are fabricated by 3D printing. The mass-loaded membrane is cut to the same circular shape as the cylindrical shell and mounted on the top of the shell. The detailed parameters of the designed devices and their corresponding eigenfrequencies can be found in Supplementary Table S1.”

[4] Sun, L., Au-Yeung, K. Y., Yang, M., Tang, S. T., Yang, Z., Sheng, P. Membrane-type resonator as an effective miniaturized tuned vibration mass damper. *AIP Advances* 6, 085212 (2016).

[5] Li, J., Chan, C. T. Double-negative acoustic metamaterial. *Physical Review E* 70, 055602(R) (2004).

[6] Peng, H., Pai, P. F. Acoustic metamaterial plates for elastic wave absorption and structural vibration suppression. *International Journal of Mechanical Science* 89, 350 (2014).

Comment 2:

• *The authors need to provide more information here and experimental testing results on the mechanical properties of this material.*

Answer: Thank you for the advice. The membrane is a commercial rubber film. The manufacturer has provided a detailed list of its mechanical properties and their testing conditions. We have added more information about the membrane's mechanical properties to the revised manuscript. The following content is added in the *Materials and Methods* section of the revised manuscript:

“The membrane of the device is made of a thin natural rubber film (McMaster-Carr Supply Company, Super-Stretchable Abrasion-Resistant Natural Rubber, 8611K11⁶¹) with a thickness of 0.1524 mm. The Young's modulus of the film within 100% elongation is 0.83 MPa and the tear strength is 1.79 MPa. The film can be elongated 750% at break with an ultimate tensile strength of 27.58 MPa.”

Comment 3:

• *Is this metamaterial sensitive to the environment?*

Answer: The membrane is a commercial rubber film. The properties related to our device, including density, Young's modulus, Poisson's ratio, tensile strength, and elongation, have negligible changes over the typical environmental conditions, such as temperature ^[7] in the range of 0 °C to 50 °C and humidity ^[8] in the range of 30 %RH to 50 %RH, that will be encountered in most of the application scenarios such as healthcare, architecture health diagnosis, and manufacturing process monitoring. However, under some extreme conditions, the rubber film will lose its functions. For example, the film will have aggravated performance decline outside the range of -60 °C to 100 °C. The water molecules in the environment will accelerate the aging of rubber film, making the device not suitable to work in high humidity.

We have added the above descriptions to the “Robustness of the device” section of the manuscript on page 9.

[7] Kaiser, S., Rabbani, R. Ahmed, R., Kaiser S., Temperature dependent mechanical properties of natural and synthetic rubber in practical structures. *Acta Mechanica Slovaca* 25, 6–14 (2021).

[8] Chang, H., Wan, Z., Chen, X., Wan, J., Luo, L., Zhang, H., Shu, S., Tu, Z., Temperature and humidity effect on aging of silicone rubbers as sealing materials for proton exchange membrane fuel cell applications. *Applied Thermal Engineering* 104, 472–478 (2016).

Comment 4:

• *It is surprising that the sensor works in all angles, as the stiffness of the metamaterial is far from rigid, the reviewer is guessing. I understand that the sensor can work in 0 and 180 degrees, but not for others. The authors need to elaborate more into this and explain why this is happening.*

Answer: The angle of the device will change the direction of the gravity force on the loaded mass with respect to the membrane plane. However, this force does not break the resonance state of our metamaterial device, which are verified by both simulation and experiment.

Fig. R3 (a) Simulation setup of the added in-plane and out-plane force and (b) Simulated input and output vibration waveforms when the input frequency is 9.5 Hz measured from the device with 9.5 Hz eigenfrequency.

We performed two additional simulations using the solid mechanics module in COMSOL Multiphysics. Besides the membrane (200 mm diameter, 0.15 mm thickness) and loaded mass (23.285 g), a body force (9.8 N/kg) with a different direction in each simulation is added to the system to simulate the rotation caused gravity direction change. The added force will offset the mass by a small distance and change the static stress distribution on the membrane. However, the results only showed a small and negligible difference between the output vibration amplitudes in the two simulations. (Fig. R3).

Fig. R3 Snapshot of designed device vibration video with device rotation of (a) 0 degree and (b) 90 degrees.

This is also proved by our laser-vibrometer measurements in Fig. 2(e) and the two newly added videos displaying the membrane vibration when the device is at 0° and 90° , respectively. We do not observe obvious change in the vibration behavior at different angles.

We have revised the manuscript on page 8:

“With the same input excitation signal, the measured displacement of the top membrane layer is similar for different inclination angles because the influence of the different gravity force angles is small and did not break the resonance state of the system, showing that the device is robust under inclination. Besides the experiments (shown in Video S1 and Video S2), we have performed numerical simulation to verify that the change in the displacement of the membrane is negligible under different inclination angles (Fig. S2 in the Supplementary Materials).”

The simulation results are added to the revised Supplementary Materials, and the videos are also attached.

Comment 5:

• The reviewer does not see why this sensor would be used for Wi-Fi identification. Aren't there much cheaper and easier ways to identify Wi-Fi? Why is this sensor important for that task?

Answer: The authors want to clarify that our device is not used for identifying the Wi-Fi signals emitted by different routers or cellphones, but for identifying the sources of mechanical vibration using reflected Wi-Fi signals from the vibrating surfaces based on the principle of Doppler effect. The Wi-Fi signals directly emitted by routers/cellphones does not allow the measurement of the vibration. It is also challenging for existing Wi-Fi-based vibration detection technologies to distinguish vibration sources, because widely used Wi-Fi antennas in routers and cellphones lack the needed directivity. Our device provides another direction to solve this problem by assigning a unique eigenfrequency for each vibration source. We have revised the descriptions in the section “Improved sensitivity and labeling effect in wireless Wi-Fi vibration sensing” on page 13:

“However, current Wi-Fi vibration sensing methods are not widely used in practice due to their limited sensitivity. First, it lacks sensitivity to small vibration amplitudes because of the limitation posed by the large wavelengths of Wi-Fi signals. Second, it is unable to differentiate different objects because widely used Wi-Fi antennas in routers and cellphones lack the needed directivity. In our proposed system, our passive vibration processing device aims to overcome the sensitivity limitation of Wi-Fi based sensing by amplifying the vibration signal and in turn its impact on CSI measurement at the Wi-Fi receiver. The ability of our proposed device to assign a unique eigenfrequency to identify an object in an environment addresses the second limitation; thus, we leverage the benefits of low-cost, non-contact Wi-Fi based sensing, as well as overcome its limitations using our novel design of the device.”

Comment 6:

• The reviewer would like to see more applications on structural health monitoring and how could this technology be applied on an actual structure, given the connectivity to a surface challenge. Also, the authors mention that these sensors could work collaboratively to provide more info. Can the authors be more specific on this topic? Examples would be preferred here.

Answer: Our device does need a surface to be attached to. However, there is no need for power cables or data transfer wires, making it more flexible compared to wired devices such as strain gauges. Our device can be attached to a vibrating surface using glue, tape, magnets, or other methods, as long as the bonding between the device and the vibrating surface is stable. Furthermore, the vibrating surface does not need to be flat. We can edit the bottom surface of the device to match that of the vibrating source. These flexibilities in the surface types and attaching methods, as well as the small footprint of our device, enable many structural health monitoring applications, such as building wall integrity and machine internal structure inspections.

When applying multiple devices for collective sensing, devices with different eigenfrequencies will be attached to different positions on the structure of interest. When the structure vibrates, we can measure the vibration motions at different positions. We will be able to process the position-dependent vibration signals to sense not only the presence of structure damage but also their locations.

We have added these descriptions to the “Conclusion” section of the manuscript:

“Our device can be attached to a vibrating surface using glue, tape, magnets, or other methods, as long as the bonding between the device and the vibrating surface is stable. Furthermore, the vibrating surface does not need to be flat. We can edit the bottom surface of the device to match that of the vibrating source.”

“For example, in structure health monitoring applications, multiple devices can be attached to different positions of the structure to sense the position-dependent vibration modes inside the structure remotely. The data can be used to monitor not only the presence of structure damages but also their locations.”

REVIEWERS' COMMENTS:

Reviewer #1 (Remarks to the Author):

None. Manuscript acceptable for publication.

Reviewer #2 (Remarks to the Author):

The revisions made have effectively addressed my comments/suggestions. I recommend accepting the manuscript for publication as it is.